# The Effect of Fabrication Error on the Performance of Mid-Infrared Metalens with Large Field-of-View

**DOI:** 10.3390/nano13030440

**Published:** 2023-01-21

**Authors:** Aoling Li, Jianhua Li, Honghui Jia, Huigao Duan, Yueqiang Hu

**Affiliations:** 1National Engineering Research Center for High-Efficiency Grinding, College of Mechanical and Vehicle Engineering, Hunan University, Changsha 410082, China; 2National Key Laboratory of Science and Technology on Test Physics & Numerical Mathematical, Beijing 100076, China; 3Greater Bay Area Institute for Innovation, Hunan University, Guangzhou 511300, China; 4Advanced Manufacturing Laboratory of Micro-Nano Optical Devices, Shenzhen Research Institute, Hunan University, Shenzhen 518000, China

**Keywords:** large field-of-view, doublet metalens, fabrication error

## Abstract

Mid-infrared large field-of-view (FOV) imaging optics play a vital role in infrared imaging and detection. The metalens, which is composed of subwavelength-arrayed structures, provides a new possibility for the miniaturization of large FOV imaging systems. However, the inaccuracy during fabrication is the main obstacle to developing practical uses for metalenses. Here, we introduce the principle and method of designing a large FOV doublet metalens at the mid-infrared band. Then, the quantitative relationship between the fabrication error and the performance of the doublet metalens with a large FOV from four different fabrication errors is explored by using the finite-difference time-domain method. The simulation results show that the inclined sidewall error has the greatest impact on the focusing performance, and the interlayer alignment error deforms the focusing beam and affects the focusing performance, while the spacer thickness error has almost no impact on the performance. The contents discussed in this paper can help manufacturers determine the allowable processing error range of the large FOV doublet metalens and the priority level for optimizing the process, which is of significance.

## 1. Introduction

The mid-infrared band (2.5–25 μm), which contains two atmospheric windows and a molecular “fingerprint” region, plays an important role in the medical and scientific fields, including in applications such as biological imaging, target detection and microscopic imaging, and imaging systems with a large field-of-view (FOV) are crucial in the above applications [1,2,3]. In large FOV imaging systems, the correction of off-axis aberrations (coma, astigmatism, field curvature, and distortion) is the key to obtaining high quality imaging; among these aberrations, the distortion causing geometric deformation can be eliminated by subsequent image processing [4,5]. The off-axis aberration correction is usually achieved by the combination of multiple lenses or lens arrays in traditional large FOV imaging systems, resulting in a large volume and heavy weight of the optical system.

Metasurface, an optical device composed of a subwavelength scale and anisotropic or isotropic scatterers (micro-nano structures) arrayed on a substrate at sub-wavelength intervals provides a possible solution to realize lightweight and compact optical components. According to the generalized Snell’s law of optics, by changing the parameters (shape, size, azimuth, etc.) of these micro-nano structures and the contrast of refractive index between them and the surrounding medium, the metasurface can flexibly adjust and control almost any electromagnetic wave parameters (phase, amplitude, polarization, frequency) [6,7,8,9]. In the application of the metasurface, the metalens is a metasurface with a lens function. It realizes the performance and function of the traditional lens while possessing the characteristics of being extremely thin and having easy integration [10,11,12,13], which is undoubtedly one of the most exciting and important features in research fields. Based on the above advantages, large FOV imaging devices based on a metalens have also been widely studied. One of the main approaches is to integrate two layers of metalens with different phase distributions on both sides of the same substrate [14,15]. The first layer of metalens is used to correct the spherical aberration and separate the normal incidence and oblique incidence rays so that they can be focused by different parts of the second layer of metalens. Different from a single-layer metalens [16,17], this construction can simultaneously correct spherical aberration and off-axis aberrations and is suitable for large numerical aperture (NA) application scenarios.

During the design of any metalens, it is common to simulate and verify its phase, performance, or function. Even if the simulation results are excellent, the devices prepared usually do not exhibit equally good performance on account of the existence of processing errors and may even differ greatly [18,19]. Although various micro-nano manufacturing methods have also been developed rapidly in the past few decades, such as electron beam lithography (EBL), focused ion beam (FIB), direct laser writing (LDW), and template transfer technology [20,21,22,23]. However, random factors such as time, temperature, human operation, or inherent physical mechanisms, such as the standing wave effect [24] and the proximity effect [25], will introduce certain fabrication errors and affect the final performances. These factors are the challenges faced by any lithography technology. As a consequence, the fabrication of micro-nano structures with high resolution, high precision, and a high aspect ratio is the key to taking the metalens from theories to practical applications. Compared with the visible band and the near-infrared band, the metalens working in the mid-infrared band is easier to achieve large-scale, low-cost, and mass production because of its larger characteristic size and is thus easier to pursue in practical applications. Therefore, it is necessary to explore the influence between the fabrication error and the device performance to guide the processing of a high-quality metalens in the mid-infrared band.

In this work, to analyze the influence of fabrication errors that may be introduced during the preparation process on the performance of a large FOV doublet metalens, the simulation analysis will be carried out in the mid-wave infrared from four aspects: critical dimension (CD) bias error of micro-nano structures, the sidewall inclination, the alignment between the two layers of metalens and the thickness of the substrate. In the second part of this paper, we first introduce the working principle and optimization design process of a doublet metalens with a large FOV and show the simulation performance results of the doublet metalens without error. In the third part, the XY and XZ plane light intensity distribution, modulation transfer function (MTF), and focusing efficiency are used as the performance evaluation indicators to compare the performance difference between the doublet metalens under each error condition and the no-error condition. Although the doublet metalens discussed in this paper operates in mid-infrared, the influence of each error on its performance is also a reference in other wavelength bands.

## 2. Principle and Design Method

Figure 1a is a schematic diagram of the doublet metalens, two layers of metalens with different phase distributions are integrated on both sides of the same substrate. Since the first layer of metalens is used to correct spherical aberration, it can be called the correction layer (CL), and the second layer of metalens is used for focusing, called the focusing layer (FL). In order to correct spherical aberration, the CL designed in this paper adopts the even polynomial as shown in Equation (1) [14,15], which has a phase distribution similar to the Schmitt plate [26] and can make the edge light diverge and the center light converge, which is contrary to the light refraction law of spherical lens. Therefore, the Schmitt plate can play a certain role in correcting spherical aberration.
(1)$$\varphi \left(x,\;y\right)=M\overset{5}{\underset{n=1}{\sum }}a_{n}{\left(\frac{r}{R_{0}}\right)}^{2n}$$
where M is the diffraction order, $r=\sqrt{x^{2}{+y}^{2}}$ is the actual radial distance from each micro-nano structure to the center of the metalens, $R_{0}$ is the normalized radius of the metalens, $a_{n}$ is the optimization coefficient, the subscript n represents the number of optimization coefficients, and the degree here is 5.

The second layer of the doublet metalens also adopts the phase profile formula as shown in Formula (1), and the off-axis aberration can be eliminated to some extent by optimizing its optimization coefficients and the thickness of the substrate [27]. In this paper, the optimization design process of the large FOV doublet metalens is jointly completed by the optical simulation software ZEMAX and the 3D finite-difference time-domain (FDTD) method. The design steps are as follows: ray tracing is carried out through ZEMAX to determine the ideal phase distribution of the CL, the thickness of the substrate, the ideal phase distribution, and the diameter of the FL; the FDTD Solutions is used to calculate the phase corresponding to the micro-nano unit structures with different size parameters to obtain the phase database; through MATLAB, phase matching is carried out at each coordinate of the metalens, where the difference between the ideal phase at the coordinate and the corresponding phase of the actual matching structure should be a minimum; verifying the final focusing performance through FDTD simulation. Before using ZEMAX for ray tracing optimization, we first determined that the doublet metalens substrate is BaF2, which is a low refractive index material commonly used in infrared, and the metalens layer (layers of Binary 2) is made of Si, a high refractive index material, and the high refractive index contrast of the two materials could enhance the phase modulation of micro-nano structures [13]. Figure 1b displays the basic dimensions of the doublet metalens optimized by ZEMAX, where the diameters of the CL and the FL are 40 μm and 74 μm respectively, the thickness of the substrate is 58 μm, FOV is 40°. Figure 1c,d shows the ideal phase distribution of the doublet metalens. It can be seen from the figure that the phase coverage of the CL is small because its role is correcting aberration, while the FL is used for focusing so its phase should cover 2π. Figure 1e shows the MTF curves of the ideal doublet metalens obtained by ZEMAX optimization at each FOV and the diffraction limit, where the MTF at each FOV is consistent with the diffraction limit.

Before using the FDTD to simulate the actual diffraction process of the doublet metalens, it is established that both the CL and the FL adopt the propagation phase design, which is the most suitable phase mechanism for wide-angle incidence [28], and it is not affected by the polarization state of the incident light, thus it has a wide range of application scenarios. The micro-nano structures adopt a cylindrical shape as shown in Figure 2a, and the introduced phase can be expressed as $\varphi =\frac{2\pi }{\lambda }n_{\mathrm{eff}}H$, where λ is the design wavelength, H is the height of the micro-nano pillar, $n_{\mathrm{eff}}$ is the effective refractive index of the micro-nano pillar, and the $n_{\mathrm{eff}}$ can be adjusted to obtain different phase responses by changing the diameter of the micro-nano pillar [29,30]. In order to accurately simulate the diffraction process and shorten the simulation time, the CL and the FL will be simulated separately, in which the propagation between the two layers of the doublet metalens and the propagation between the FL and the focal plane are calculated by far-field projection. Based on this, the no-error doublet metalens is simulated at a wavelength of 4.2 μm. According to the Nyquist sampling theorem, the unit period of micro-nano structures is determined to be 2 μm and the diameter varies from 0.5 μm to 1.5 μm. Figure 2b is the phase cloud diagram of the phase of the micro-nano structure changing with its height and diameter, based on this, the micro-nano structure height of the CL and the FL is determined to be 0.7 μm and 2.3 μm respectively by weighing the phase matching error and transmission. Figure 2c,d shows the comparison between the ideal phase and the actual phase of the CL and the FL, the abscissa is the radial distance to the center of the metalens. From the figure, it can be seen that the actual phase of the doublet metalens matches the ideal phase to a high degree.

Figure 3a displays the light intensity distribution along the actual focal plane and the focal length direction of the above no-error doublet metalens at three incidence angles of 0°, 10°, and 20°, which were obtained through FDTD simulation. During the simulation process, the ring PEC (Perfect Electrical Conductor) material is used to act as the aperture stop, it is worth mentioning that through our simulation, it is found that the diffraction effect introduced by PEC ring itself is negligible, which means that its introduction has no impact on the realization of our system functions. The actual focal plane obtained by simulation is located at Z = 38 μm, which is a 2 μm difference from the design focal length, and the possible reason is that the metalens acts as a plane in ZEMAX, but when we use FDTD Solutions to simulate the diffraction process of the metalens, the metalens has a certain height so that the actual value and the design value have a certain deviation. Figure 3b shows the MTF curves obtained by the Fourier transform of the intensity distribution at the focal plane, which comprehensively reflects the resolution and contrast of the metalens [31,32], and the larger the coverage area under the MTF curve, the better the metalens. It can be seen from the figure that compared with the diffraction limit, the MTF curve at each FOV has a certain attenuation, which is caused by aberration and diffraction. Figure 3c shows the corresponding focusing efficiency, where the maximum focusing efficiency is 75.50%, the average focusing efficiency is 68.14%, and the focusing efficiency here is defined as the light intensity in an area of diameter 3 × FWHM (full width at half-maximum) on the focal plane divided by the total incident light intensity [29].

In the process of manufacturing the metalens, errors will inevitably be introduced. For example, during the photolithography process, some accidental factors affect the manufacturing quality, such as the thickness of the photoresist, mechanical vibration, ash particles, temperature changes, development time, pretreatment conditions and so on, which can be solved through multiple process experiments and the optimization of process parameters. However, it is difficult to solve the problems caused by inherent physical mechanisms such as the standing wave effect and the proximity effect. At present, most of the high aspect ratio etching processes use inductively coupled plasma reactive ion etching (ICP-RIE) [33], which has the advantages of high control accuracy, large area etching consistency, good vertical characteristics and low material loss, etc. In the field of deep silicon etching, the currently widely adopted process flow is based on the etching-passivation gas alternation technology invented by Bosch [34,35], that is, a passivation layer is generated on the side wall of the etched material, and the etching gas will etch both Si and the passivation layer at the same time, which can protect the Si of the sidewall from lateral etching, but attention needs to be paid to controlling the time interval between the passivation and the etching. In the etching process, in addition to the above accidental factors, there are also some factors affecting the etching process, such as the selection and proportion of reaction gas, working pressure in the cavity, and inductively coupled plasma (ICP) power, all of which also affect the structure morphology and size.

## 3. Error Analysis and Simulation Results

### 3.1. Critical Dimension (CD) Bias Error

CD bias error refers to the inconsistency between the radial size of the fabricated micro-nano structures and the ideal design, which is a common error in nanomanufacturing technology. Most photoresist morphology is regular trapezoidal, and the photoresist at the edge is relatively thin, so the glue withdrawal phenomenon will occur during the etching, that is, the photoresist in the edge area is exhausted, the radial size of the pillars will become smaller and thinner. At the same time, in the Bosch process, in order to maintain anisotropic etching, a balance must be achieved between the etching agent and the passivator agent. If the balance is broken due to too much etching agent, the process will become more isotropic, resulting in CD loss. In the alternating cycle process of etching-passivation, the rough sidewall is inevitable [34,36]. Since the roughness is regular and occurs on a length scale much smaller than the wavelength, this error can be simplified as the reduction of the diameter of the micro-nano pillars to facilitate qualitative analysis. In this paper, the CD of micro-nano structures is reduced as a whole. Figure 4a shows a schematic of CD bias error, where M is the reduction coefficient. Based on this, we simulate and analyze the situation when the M coefficient is 95.0%, 90.0%, 85.0%, 80.0%, 75.0%, and 70.0%. The focusing efficiency and MTF curves obtained under each error condition are obtained under the respective actual focal plane, rather than the focal length under the no-error condition. Therefore, under some specific errors, one or two performances could be better than that under the no-error condition. Figure 4b,c shows the light intensity distribution along the XY and XZ directions at different FOVs under various error conditions. It can be found that with the increase of CD bias error, the sub-diffraction order is more obvious. At the same time, it can be discovered in the MTF curves as shown in Figure 4d that when the error is 80.0% and 70.0%, the MTF curves fluctuate, indicating that there exists field curvature. In order to facilitate the readers to distinguish the MTF curves under different errors and FOVs, this paper only selectively displays the MTF curves under partial errors. Besides, this paper also extracts the MTF values of each MTF curve at the frequency of 33.33 lp/mm and draws a broken line chart as shown in Figure 4e for the convenience of readers’ comparison. From the chart, when the error increases from 100.0% to 70.0%, the average MTF value decreases from 0.716 to 0.154; meanwhile, Figure 4f shows that the average focusing efficiency reduces from 68.14% to 33.41%. In addition, we can discover that the focusing efficiency is higher when the errors are 95.0% and 90.0% than when there is no error, mainly due to the increase of transmittance of the FL, the essence is: the diameters of the micro-nano structures are reduced as a whole, the spacing between structures increases, that is, the coupling between structures decreases so that the phase modulation capacity and transmittance can be optimized to a certain extent, this phenomenon occurs within a certain CD reduction range.

### 3.2. Inclined Sidewall Error

The inclined sidewall error is that the radius of the upper surface and the lower surface of the micro-nano structure is not consistent, resulting in a slope-like sidewall. In addition to the accidental factors, the anisotropy of the process itself will also cause the inclined sidewall error. Its essence is: deep reactive ion etching (DRIE) is performed by continuous alternating cycles of the passivation and the etching to achieve the desired etching depth, but as the etching depth continues to increase, the active F-base density reaching the bottom will gradually become smaller, and the etching speed will also reduce so that the etching process and the passivation process cannot reach an effective balance, which will affect the sidewall roughness and verticality [37,38]. Although this effect can be reduced by controlling the alternating intervals of the etching process and the passivation process, the existence of an inclined sidewall cannot be avoided. To analyze its influence on the large FOV doublet metalens, in this paper, the same inclination is applied to all micro-nano structures when simulating a single error condition. Figure 5a is a schematic diagram of the inclined sidewall error, and Figure 5b–f shows the focusing performance results of the doublet metalens under the six cone angles of 1.0°, 2.5°, 5.0°, 7.5°, 10.0°, and 12.5°. As can be seen from Figure 5c, the sub-diffraction order is obvious from the cone angle of 7.5°, especially when the cone angle is 12.5°. This law is also reflected in Figure 5b. At the same time, from Figure 5d, the MTF curves under large error conditions presented have fluctuations, that is, there have field curvature, it also can be manifested from Figure 5c that the focal lengths of the three FOVs are not in the same plane under large error conditions. From Figure 5e,f, when the cone angle increases from 0° to 12.5°, the average MTF value at the frequency of 33.33 lp/mm decreases from 0.716 to 0.155, and the average focusing efficiency decreases from 68.14% to 19.64%. The reduction of focusing efficiency is mainly reflected in two aspects. One is the reduction of transmittance of the FL. Figure 5e presents the broken line chart of the transmittance of the selected structure of the FL changing with its bottom radius. With the increase of the cone angle, there are more pillars that possess low transmittance, leading to the reduction of the transmittance of the FL. The other aspect is the decrease of the relative focusing efficiency on the focal plane. It can be seen from Figure 5h that when the cone angle increases to a certain extent, the phase provided by the structures cannot cover 2π and eventually will be unable to focus.

### 3.3. Interlayer Alignment Error

The interlayer alignment error refers to the deviation between the center position of the currently prepared metalens pattern and the center position of the already prepared metalens pattern on the substrate. The causes of such errors include the deformation of the wafer itself, the error caused by the lithography machine, the uneven movement of the wafer workpiece, the error of the alignment mark and the environmental factors [39]. This process is difficult, and the prepared pattern should also be protected from destruction when processing the second layer of patterns. In order to quantitatively analyze the interlayer alignment error, this paper takes the center of the CL as the standard, and offsets the center of the FL in the X direction. Figure 6a is a schematic diagram of the interlayer alignment error, and ∆S represents its offset. In this paper, we simulate the different doublet metalens when ∆S is 5.0%, 10.0%, 15.0%, 20.0%, 25.0% and 30.0% of the diameter of the FL (D_FL_), respectively. From Figure 6b,c, it can be seen that as the error increases, the focusing beam is deformed and the focusing performance degrades. From Figure 6d,f, when the error increases from 0.0% to 30.0%, the average MTF value at the frequency of 33.33 lp/mm decreases from 0.716 to 0.587, and the average focusing efficiency decreases from 68.14% to 43.88%.

### 3.4. Spacer Thickness Error

Spacer thickness error is the difference between the actual thickness of the substrate and the ideal thickness. The spacer thickness in this paper is obtained by ZEMAX optimization, if we want the actual thickness of the substrate to be consistent with this value, a grinding process is required. The wafer grinding process, also known as the backside grinding process, is the process of thinning the back side of the wafer. Grinding the wafer thickness to the ideal range is one of its goals. The wafer grinding error is inversely proportional to the wafer thickness [40], so the quality of the wafer can be improved if the grinding thickness can be minimized. One of the objectives of this paper is to find the tolerable range of the spacer thickness error. Figure 7a shows the spacer thickness error schematic diagram, where the ∆L represents the spacer thickness increment. This paper simulates the doublet metalens under the situation when ∆L is 5.0%, 10.0%, 15.0%, 20.0%, 25.0%, and 30.0% of the ideal spacer thickness L, respectively. As can be seen from the light intensity distribution diagrams in XY direction and XZ direction in Figure 7b,c, even if the error increases to the maximum, it only has a slight influence on the performance of the doublet metalens. From the MTF curves and the broken line chart of focusing efficiency corresponding to Figure 7d–f, when the error increases from 0.0% to 30.0%, the average MTF value changes slightly from 0.716 to 0.729, and the average focusing efficiency changes from 68.14% to 68.57%. It can be seen that the discrepancy of the spacer thickness has little effect on the focusing performance of the doublet metalens.

## 4. Conclusions

In summary, the basic design principle and method of a mid-infrared doublet metalens with a large FOV are introduced, and the effects of common errors in micro-nano fabrication on the performance of the doublet metalens are simulated and analyzed, including CD bias error, inclined sidewall error, interlayer alignment error, and spacer thickness error. The simulation results show that both the CD bias error and the inclined sidewall error have a great impact on the focusing performance, and the influence of the inclined sidewall error is more serious. Under the situation of inclined sidewall error, MTF decreases rapidly and fluctuates obviously, and the focusing efficiency also decreases significantly with the increase of the error. As for the interlayer alignment error, it not only deforms the focusing beam but also affects the final focusing efficiency. At the same time, the effect of spacer thickness error on the focusing performance is minimal, and the effects caused by this error are negligible according to the performance analysis results. The influence between fabrication error and the performance of large FOV doublet metalens is explored in this paper, which can help manufacturers determine the allowable processing error range to guide the processing of high-quality doublet metalens.

## Figures and Tables

**Figure 1 nanomaterials-13-00440-f001:**
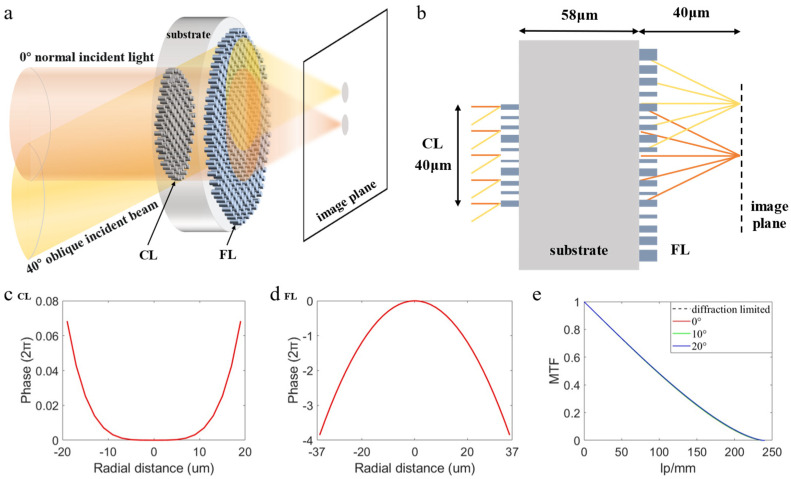
(**a**) Schematic diagram of the large FOV doublet metalens. (**b**) Schematic diagram of basic dimensions of the large FOV doublet metalens. (**c**,**d**) Ideal phase distribution of the CL and the FL. (**e**) MTF curves at each FOV and the diffraction limit.

**Figure 2 nanomaterials-13-00440-f002:**
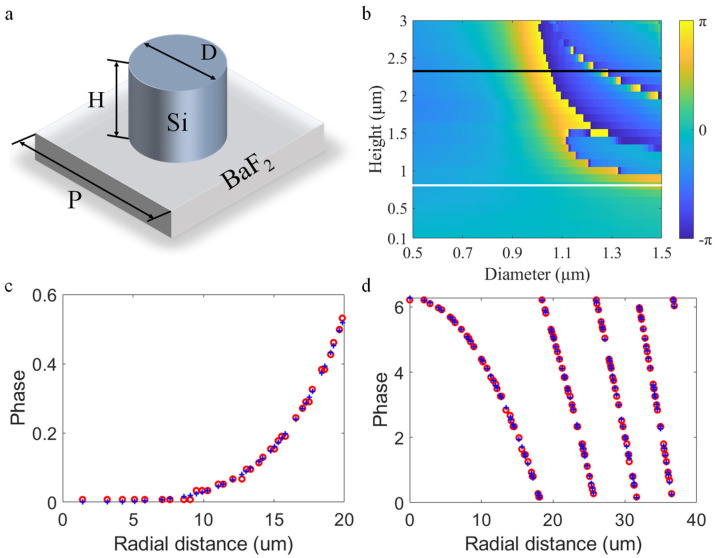
(**a**) Schematic diagram of micro−nano unit structure. (**b**) Simulated phase shift as a function of micro−nano pillar radius and height, the white and the black line represents the micro−nano structure height of the CL and the FL, respectively. (**c**,**d**) Ideal phase distribution of the CL and the FL, the blue plus and circle red markers represent the ideal and design phase, respectively.

**Figure 3 nanomaterials-13-00440-f003:**
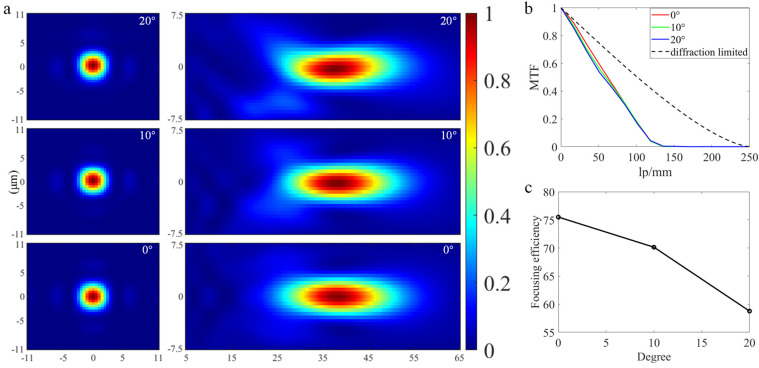
(**a**) Light intensity distribution along XY and XZ direction at each FOV. (**b**) MTF curves at each FOV and the diffraction limit. (**c**) Focusing efficiency at 0°, 10°, and 20° FOV.

**Figure 4 nanomaterials-13-00440-f004:**
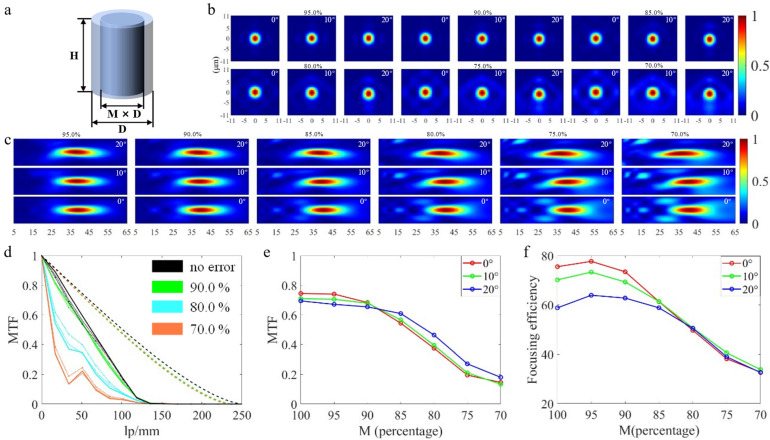
(**a**) Schematic diagram of CD bias error. (**b**,**c**) Light intensity distribution along XY and XZ direction at different FOV under various error conditions. (**d**) MTF curves at different FOV under various error and no−error conditions, the double dash line represents the diffraction limit, the solid line represents the 0° FOV, the dash line represents the 10° FOV, the dot−dashed line represents the 20° FOV (the diffraction limit at 80.0% error coincides with that at 70.0% error). (**e**) Line chart of MTF values at different FOV under various error and the no−error conditions when the frequency is 33.33 lp/mm. (**f**) Focusing efficiency at different FOV under various error and no−error conditions.

**Figure 5 nanomaterials-13-00440-f005:**
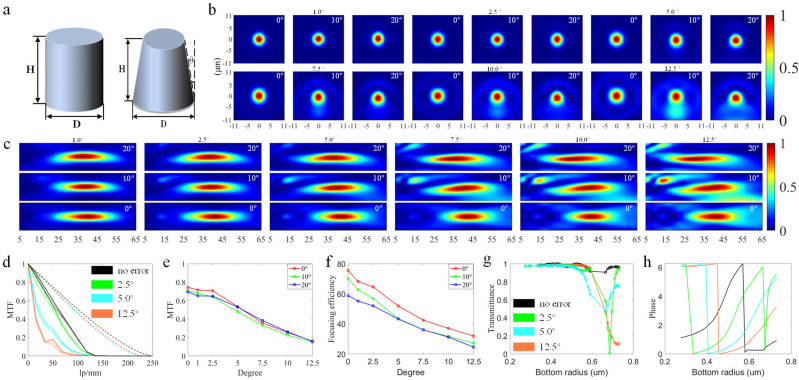
(**a**) Schematic diagram of inclined sidewall error. (**b**,**c**) Light intensity distribution along XY and XZ direction at different FOV under various error conditions. (**d**) MTF curves at different FOV under various error and no−error conditions, the double dash line represents the diffraction limit, the solid line represents the 0° FOV, the dashed line represents the 10° FOV, the dot−dashed line represents the 20° FOV (the diffraction limit at 2.5° error coincides with that at 12.5° error). (**e**) Line chart of MTF values at different FOV under various error and no−error conditions when the frequency is 33.33 lp/mm. (**f**–**h**) Focusing efficiency at different FOV, transmittance with different bottom radius of pillars, phase response with different bottom radius of pillars under various error and no−error conditions.

**Figure 6 nanomaterials-13-00440-f006:**
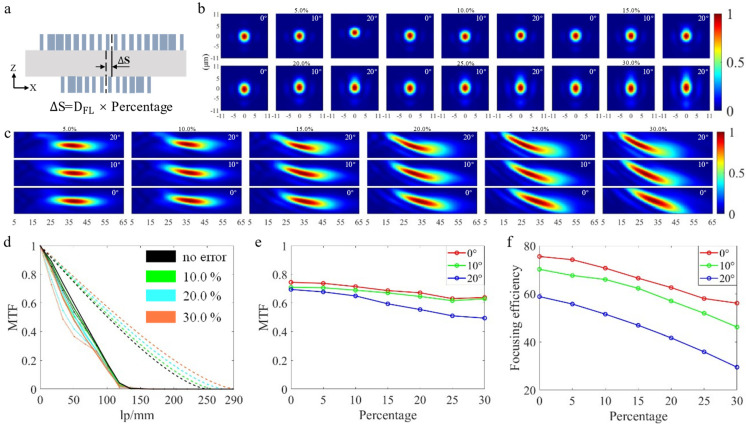
(**a**) Schematic diagram of interlayer alignment error. (**b**,**c**) Light intensity distribution along XY and XZ direction at different FOVs under various error conditions. (**d**) MTF curves at different FOV under various error and no−error conditions; the double dashed line represents the diffraction limit, the solid line represents the 0° FOVs; the dashed line represents the 10° FOVs; the dot−dashed line represents the 20° FOV. (**e**) Line chart of MTF values at different FOV under various error and no−error conditions, when the frequency is 33.33 lp/mm. (**f**) Focusing efficiency at different FOV under various error and no−error conditions.

**Figure 7 nanomaterials-13-00440-f007:**
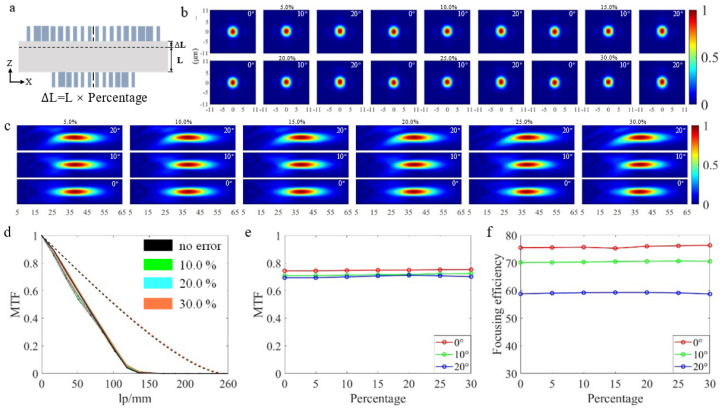
(**a**) Schematic diagram of spacer thickness error. (**b**,**c**) Light intensity distribution along XY and XZ direction at different FOV under various error conditions. (**d**) MTF curves at different FOVs under the conditions of various errors and no−error, the double dashed line represents the diffraction limit, the solid line represents the 0° FOV; the dashed line represents the 10° FOV, the dot−dashed line represents the 20° FOV (the diffraction limit under each error condition coincide with the no−error one). (**e**) Line chart of MTF values at different FOV under various error and no-error conditions when the frequency is 33.33 lp/mm. (**f**) Focusing efficiency at different FOV under various error and no−error conditions.

## Data Availability

The data presented in this study are available on request from the corresponding author.

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
