# Peer review of "The Effect of Fabrication Error on the Performance of Mid-Infrared Metalens with Large Field-of-View"

_nanomaterials, 2023, doi:10.3390/nano13030440_

Round 1
Reviewer 1 Report
The paper by Aoling Li, Jianhua Li, Honghui Jia, Huigao Duan, and Yueqiang Hu is devoted to the computer design of a mid-infrared double metalense. The design itself is well known however, the authors thoroughly investigate how the errors in design and/or fabrication are influence the imaging quality. The design principle and method of a metalense with large field-of-view are introduced. The metalense is composed, of periodical arranged cylindrical elements that locally change the phase. Effects of common errors in micro-nano fabrication are simulated and analyzed, including, inclined sidewall erroring in a cylindrical element, interlayer alignment error and spacer thickness. The simulation results show that the inclined sidewall error have a great impact on the focusing performance. Under the situation of inclined sidewall error, modulation transfer function decreases rapidly and fluctuates obviously, and the focusing efficiency decreases significantly with the increase of the error. As for the interlayer alignment error, it not only deforms the focusing beam, but also affects the final focusing efficiency. At the same time, the effect of spacer thickness error on the focusing performance is minimal, and the effects caused by this error are negligible according to the performance analysis results.
The influence between fabrication error and the performance of large field-of-view doublet metalense is explored in the paper, which can help manufacturers determine the allowable processing error range. It is an interesting paper that can be published.
Yet, I have a comment. The authors designed the metalense from the cylindrical elements. Then they found that the replacement of the original elements by cone frustums much decreases the focusing efficiency. Could the metalense be made of the cone frustums from the beginning?
Reviewer 2 Report
This paper describes a numerical analysis of a focusing performance of a doublet metalens for a large FOV. A quantitative assessment of fabrication error was performed and concluded that the CD bias and incline sidewall errors significantly affect the lens performance. I believe that this result has an enormous impact on the audience, especially those fabricating metalenses. If a few concerns below are successfully revised, I should agree to the publication of this paper on Nanomaterials.
1. In line 134, please define H and lambda.
2. In Fig. 2(c) and (d), which is ideal and design for plus and circle markers?
3. In Fig. 2(c) and (d), the phase values of red circle markers seem to be quantized. Why?
4. In line 154 or around the first sentence of this paragraph, it should be mentioned that the result of Figure 3 is obtained through FDTD for the reader's understanding.
5. Lines 160-164, descriptions of Fig. 3b and 3c are reversed.
6. For the reader's understanding, I think it would be better to add other diagrams of meta-atom properties (transmittance, phase difference v.s. width), especially including the inclined sidewalls.
7. In Fig. 6d, diffraction-limited performance becomes better when the alignment error is large. Why?
8. There are some English errors for example (but not limited) as follows.
* L78 A period is missing after "substrate".
* L99 number -> degree
* L123 bimetal -> doublet
